# Psychosocial and socio-environmental factors associated with adolescents' tobacco and other substance use in Bangladesh

Md. Mostaured Ali Khan[1,2☯], Md. Mosfequr Rahman[1☯]*, Syeda S. Jeamin[3], Md. Golam Mustagir[1], Md. Rajwanul Haque[2], Md. Sharif Kaikobad[1]

1 Department of Population Science and Human Resource Development, University of Rajshahi, Rajshahi, Bangladesh, 2 MEL and Research, Practical Action, Dhanmondi, Dhaka, Bangladesh, 3 Department of Sociology and Psychology, University of North Texas at Dallas, Dallas, TX, United States of America

☯ These authors contributed equally to this work.
* mosfeque@ru.ac.bd

## Abstract

### Background

Tobacco, alcohol, and marijuana are the top three substances used by adolescents. The adverse health effects of these three substances are well documented in epidemiological literature, yet little is known about the substance use and associated factors among adolescents in Bangladesh. This study examines the risk factors for tobacco and other substances use among school-going adolescents in Bangladesh.

### Methods

We analyzed data from the 2014 Bangladesh Global School-based Student Health Survey (GSHS) of adolescents aged 13–17 years. We used two outcome measures: tobacco use (TU) and other substance use (SU; alcohol and/or marijuana). We examined a set of reported psychosocial and socio-environmental adverse events as risk factors. Logistic regression analyses were used to identify associations.

### Results

The prevalence of TU and other SU among school-going adolescents was 9.6% and 2.3%, respectively. The likelihood of TU and other SU was higher among adolescents who reported being bullied (TU: adjusted odd ratio [AOR]: 1.93; 95% confidence interval [CI]: 1.24–3.00; and other SU: AOR: 3.43; 95% CI: 1.46–7.99) and having sexual history (TU: AOR: 19.38; 95% CI: 12.43–30.21; and other SU: AOR: 5.34; 95% CI: 2.17–13.29). Moreover, anxiety-related sleep loss was associated with adolescents' TU (AOR: 2.41; 95% CI: 1.02–5.82) whereas the likelihood of other SU (AOR: 3.27; 95% CI: 1.14–9.44) was higher among lonely adolescents. Experience of adverse socio-environmental factors, such as parental substance use (TU: AOR: 7.81; 95% CI: 5.08–12.01), poor monitoring (TU: AOR: 1.96; 95% CI: 1.16–3.31) and poor understanding (TU: AOR: 2.22; 95% CI: 1.36–3.65), and lack of peer support (TU: AOR: 3.13; 95% CI: 1.84–5.31; and other SU: AOR: 2.45; 95% CI:

**Data Availability Statement:** The data underlying the results presented in the study are available

from: https://www.cdc.gov/gshs/countries/seasian/bangladesh.htm.

**Funding:** The authors received no specific funding for this work.

**Competing interests:** The authors have declared that no competing interests exist.

1.02–5.91), and truancy (other SU: AOR: 4.29; 95% CI: 1.81–10.12) were also positively associated with TU and/or other SU. Additionally, higher odds of tobacco use were observed among adolescents who reported 1 (AOR: 4.36 times; 95% CI: 1.34–14.24), 2 (AOR: 8.69 95% CI: 1.67–28.23), and ≥3 (AOR: 17.46; 95% CI: 6.20–49.23) adverse psychosocial experiences than who did not report any psychosocial events.

## Conclusions

Tobacco and other substance use among school-going adolescents are prevalent in Bangladesh. Several psychosocial and socio-environmental events are associated with TU and other SU, which should be incorporated into adolescent substance use and health promotion programs.

## Introduction

Unhealthy behaviors, such as tobacco use, drinking alcohol, and the use of illicit drugs, particularly among adolescents, are crucial global public health issues [1]. Worldwide, around 7.6% of adolescents are found as current smokers (smoked in the past 30 days), and more than a quarter (26.5%, 155 million) are current drinkers [2]. Around 35% of the total population in Bangladesh use either smoked cigarettes or *bidis* or smokeless tobacco products (tobacco flakes known as *Zarda*, betel leaf quid or *paan*, and gul [3]. In Bangladesh, tobacco use contributes to approximately 1.6 million deaths every year (19% of all deaths) [4]. The 2013 Global Youth Tobacco Survey of Bangladesh reported that 6.9% of adolescents (9.2% males and 2.8% females) were tobacco users [5]. The prevalence of alcohol drinking across age groups is less than one percent (0.8%; men 1.5% and women 0.1%) [6]. A recent report documents that more than 7 million people suffer from drug addiction in Bangladesh, of whom 25% are below 15 years old [7]. Bangladesh is a route of illicit drug trafficking as its geographical location is in between the "golden triangle" (Myanmar, Thailand, and Laos) and "golden crescent" (Pakistan, Afghanistan, and Iran), which makes the drugs available to people in all segment of the society [7].

Tobacco and other substance (which includes alcohol and/or marijuana in this study) use in a key period of transition, such as adolescence, has been the cause of increasing concern. Adolescence might be a vulnerable time for the development of substance dependence [8]. Substance use cause problems at all age levels but appears to be more dangerous in adolescence [9], contributing nearly half of the morbidities among adolescents [10]. Substance use has adverse consequences on individuals, families, and communities, which contribute to negative physical and mental health as well as social problems [11]. Specifically, it is found to be a major contributing factor for non-communicable diseases, poor health, mental illness, suicidal behaviors, and decreased life expectancy [12, 13]. Therefore, understanding the underlying factors associated with tobacco and other substance use among adolescents is essential for developing prevention strategies and to attain the global sustainable goals (SDGs) related to reducing premature mortality from non-communicable diseases, and non-intentional deaths, and injuries [14].

Several risk and protective factors for substance use among adolescents and young adults have been identified around the globe. For example, previous studies have documented several individual and interpersonal risk factors for substance use among adolescents, such as abuse or neglect [15], stressful life events [15], poor family relations [15, 16], internalizing and

externalizing behaviors [15, 17], adolescent employment [15], and lower educational attainment [18, 19]. Moreover, family relationships (eg., guidelines and monitoring) [15, 20], being married or in a stable relationship [15], are identified as protective factors for these unhealthy behaviors. Adolescents' tobacco and other substance use are also highly associated with availability and access to drugs and alcohol [15, 21], peer smoking, peer norms and attitudes [15, 22], and school factors [23, 24]. There is a large body of literature that highlights psychosocial factors such as loneliness, stress, depression, anxiety, hyperactivity/inattention, and low self-esteem as risk factors for substance use. For example, Page et. al. [25] reported in a four country study (Philippines, China, Chile, and Namibia) that among adolescents often feeling lonely, worried, sad/hopeless, or having a suicide plan were associated with current smoking and drinking alcohol. Findings from a longitudinal study in Australia suggest that psychosocial risk factors, such as truancy, hyperactivity/inattention, and behavior problems are associated with adolescent binge drinking and cannabis use [26]. A recent review concludes that psychosocial risk factors of adolescent substance use are closely related to peer influence, and that parents tend to have the strongest effect on adolescents' substance use behavior [27].

Only a few studies in Bangladesh focused to identify the factors associated with adolescents' tobacco use. For example, Kabir et al. [28] reported that friends' tobacco use, receiving pocket money, receiving free tobacco products from vendors, and exposure to advertisements and promotions of tobacco products increase the likelihood of adolescents' tobacco use. These findings are also supported by a study among adolescent in South-Asian countries [29]. Furthermore, smoking by teachers and peer influence strongly associated with smoking among secondary school going adolescents [30]. Ullah et al. [31] documented that perceived knowledge on severity of tobacco use and barrier of using tobacco are protective for adolescents' smokeless tobacco use. However, all these studies are either limited to only tobacco (smoke or smokeless) use or used rural samples only. Moreover, none of these studies have taken into account psychosocial factors (such as loneliness, bullying victimization, etc.) as the factors associated with adolescents' tobacco, alcohol or marijuana use. Therefore, how psychosocial factors, which are found to be risk factors in some global studies [25–27], are associated with tobacco and other substance use among adolescents in Bangladesh is unknown. To address this knowledge gap using nationally representative data, this study estimates the prevalence and factors associated with tobacco and other substance use (alcohol and/or marijuana) among school-going adolescents in Bangladesh, with a particular focus on adverse psychosocial and socio-environmental factors. A growing number of studies document a graded relationship between adverse life circumstances and negative health outcomes, including health risk behaviors, such as substance use [32, 33]. Therefore, we also examine to what extent multiple adverse experiences (MAE) affect tobacco and other substance use behaviors among the adolescent population. Identifying these relationships will be crucial for developing effective anti-substance use programs in Bangladesh.

## Data and methods

### Sample design

Data for current analyses were extracted from the most recent Global School-based Student Health Survey (GSHS) of Bangladesh, 2014. The GSHS is a population-based survey of school-going adolescents around the world, which is conducted by the World Health Organization (WHO) in collaboration with the Center for Disease Control and Prevention (CDC). The GSHS collects information regarding different aspects of adolescent health and behaviors with the aim to help countries in developing appropriate school and adolescent health policies and programs as well as facilitating comparison of collected information across countries. The

GSHS used a clustered sampling technique via a standardized scientific sample selection process with a conventional school-based methodology. The 2014 Bangladesh GSHS is a cross-sectional survey of a two-stage stratified cluster sampling design, conducted among students in grades 7, 8, 9, and 10. At the first stage, schools were selected based on a probability proportional to size sampling, and at the second stage, grade levels within each of those schools were selected randomly, and all students in selected classes were included in the sampling frame. Participants completed a self-administered questionnaire and recorded their responses on a computer scannable answer sheet. Participants were only allowed to use pencils that were provided to them to answer the questions. The questionnaire consisted of 80 core, expanded and country specific questions. The survey used the Bengali version of the questionnaire [34] A total of 3,180 eligible students were selected to participate; 2,989 (94%) of these students were interviewed. This study was approved by the Ministry of Health and Family Welfare and the Ministry of Education in Dhaka, Bangladesh. To ensure voluntary participation, privacy, and confidentiality, informed consent was obtained from the students, parents and/or school officials [35]. Further details of the GSHS sampling and data are available at:

https://extranet.who.int/ncdsmicrodata/index.php/catalog/485 and https://www.cdc.gov/gshs/countries/seasian/bangladesh.htm

## Outcomes

Adolescents' tobacco use (smoking cigarette and/or using other tobacco products) and other substance use (drinking alcohol and/or using marijuana) were the two primary outcomes of interest. Tobacco use (TU) was assessed by adding the response of two items: "During the past 30 days, on how many days did you smoke cigarettes?" and "During the past 30 days, on how many days did you use any tobacco products other than cigarette, such as biri, jarda, tobacco leaf, gul, or shisha?" For either item, those who responded using it for one or more days were considered to be tobacco users and coded 1, and coded 0 if they responded zero days to both the items. Other substance use (SU) was assessed combining the use of two illegal substances-alcohol and marijuana [36]. Alcohol and marijuana use were measured based on the response to the items: "During the past 30 days, on how many days did you have at least one drink containing alcohol?" and "During the past 30 days, how many times have you used marijuana (also called ganja or weed)?" Adding up the responses of these two items, other SU was recoded as: 0 = never use alcohol and/or marijuana; and 1 = drink alcohol and/or use marijuana at least one day.

## Explanatory variables

Based on reviewing prior literature [37, 38], a set of socio-demographic, psychosocial, and socio-environmental factors were used as explanatory variables in this study to examine how they are associated with adolescents' tobacco and other substance use. Categories and coding of these explanatory variables are presented in Table 1.

## Calculation of multiple adverse experience (MAE) scores

We followed the Kaiser Permanente Childhood Adverse Experience (ACE) study to calculate the multiple adverse experience (MAE) score [39]. We divided adolescents' adverse experiences into two categories: psychosocial and socio-environmental factors. Details of these two categories are presented in Table 1. The MAE score for each adverse category was calculated by summing up the responses on the adverse experience questions. For each adverse experience, the response 0 meant the participant never or rarely experienced the adversity, while 1 meant the participant always or sometimes experienced that specific adversity. Higher MAE

**Table 1.  The complete list of independent variables.**

| Variables | Survey question | Coding |
|---|---|---|
| Socio-demographic factors | | |
| **Age** | How old are you? | 11–18 years (coded categorically) |
| **Gender** | What is your sex? | 1 = Male |
| | | 2 = female |
| **Food insecurity (Proxy of socioeconomic status)** | How often did you go hungry because of there was not enough food in your home? | 0, No = Never/rarely/sometimes |
| | | 1, Yes = Most of the time/always |
| Psychosocial factors | | |
| **Loneliness** | How often have you felt lonely? | 0, No = Never/rarely/sometimes |
| **Anxiety** | How often have you been so worried about something that you could not sleep? | 1, Yes = Most of the time/always |
| **Bullied** | How many days you were bullied? | 0, No = Never |
| | | 1, Yes = One or more days |
| **No close friends** | How many close friends do you have? | 0, No = Have at least one close friend |
| | | 1, Yes = No close friends |
| **Physically abused** | How often you were physically attacked? | 0, No |
| | | 1, Yes = One or more times |
| **Sexual history** | Have you ever had sexual intercourse? | 0 = No |
| | | 1 = Yes |
| Socio-environmental factors | | |
| **People use tobacco in presence of adolescents** | Do people smoke in your presence? | 0, No |
| **Parental tobacco use** | Do any of your parents or guardians use any forms of tobacco? | 1, Yes |
| **Parents rarely check homework** | How often did your parents check to see if your homework was done? | 0, No = Most of the time/always |
| **Poor understanding with parents** | How often did your parents or guardians understand your problems and worries? | 1, Yes = Never/rarely/sometimes |
| **Poor parental monitoring** | How often your parents or guardians really know about what you were doing with your free time? | |
| **Lack of peer support** | During the past 30 days, how often were most of the students in your school kind and helpful? | 0, No = Most of the time/always |
| | | 1, Yes = Never/rarely/sometimes |
| **Truancy** | During the past 30 days, on how many days did you miss classes of school without permission? | 0 = 0 days |
| | | 1 = 1 or more days |

meant greater experience of adverse events. As for example, if an individual has experienced three adverse events, then the MAE score is 3.

## Statistical analysis

Data of this study were analyzed using STATA 14.0 SE (StataCorp. LP, College Station, TX, USA). The prevalence of tobacco and other substance use were assessed for total sample and by subgroups. Chi-squared tests were used to assess bivariate comparisons. Assessment of the

extent to which various adverse psychosocial and social-environmental factors are associated with adolescents' tobacco and other substance use were carried out using a logistic regression approach. We estimated both unadjusted and adjusted odds ratios and their 95% confidence intervals were calculated. In all statistical analyses, we set α = 0.05. Imputation using a logistic regression model was used to estimate missing values from known values to account for missing data, most frequently for sexual history (i.e., for 11.6% of respondents) [40]. Age, gender, and school grade were included as covariates in the imputation. Multicollinearity of the variables was checked using variance inflation factor (VIF), and in all cases, the values of VIF were found less than 2, indicating multicollinearity was not an issue. In all analyses, the complex survey design and sampling weights were considered.

## Results

Table 2 presents the sample characteristics and the prevalence of TU and other SU by selected characteristics. Majority of the students were male (65.3%), and 14.3% of respondents had food insecurity at home (a proxy of socioeconomic status). Among psychosocial factors, 10.9% of adolescents reported being lonely, 24.5% reported being bullied, and 9.3% of adolescents reported having sexual history. In addition, among socio-environmental factors, more than two-third (70.3%) of adolescents' parents used tobacco or drug, more than half of the adolescents had poor understanding with their parents (52.4%), and had poor parental monitoring (56.7%). Nearly one in three adolescents reported truancy (30.9%), and 44% reported lack of peer support at school (Table 2).

The prevalence of TU and other SU among this sample was 9.6% and 2.3%, respectively. The prevalence of TU and other SU were significantly higher among adolescents who reported loneliness (TU: 16.7%; 95% CI [9.9–26.8] vs. 8.4%; 95% CI [4.9–14.1]; P = 0.009; and other SU: 9.1%; 95% CI [4.4–17.4] vs. 1.4%; 95% CI [0.6–2.9]; P<0.001), anxiety-related sleep loss (TU: 20.5%; 95% CI [11.7–33.3] vs. 8.9%; 95% CI [5.5–13.9]; P = 0.012; and other SU: 14.3%; 95% CI [6.0–30.3] vs. 1.6%; 95% CI [0.9–3.0]; P<0.001), being bullied (TU: 18.4%; 95% CI [8.6–35.1] vs. 6.4%; 95% CI [4.2–9.9]; P = 0.014; and other SU: 6.3%; 95% CI [2.9–12.8] vs. 0.9%; 95% CI [0.5–1.8]; P = 0.001) and having sexual history (TU: 47.2%; 95% CI [31.1–64.1] vs. 5.7%; 95% CI [2.8–11.3]; P<0.001; and other SU: 12.6%; 95% CI [4.9–28.7] vs. 1.2%; 95% CI [0.5–2.5]; P<0.001) than their respective counterparts. Similarly, the prevalence of TU (19.3%; 95% CI [10.9–32.1] vs. 1.1%; 95% CI [0.3–3.7]; P = 0.024) and other SU (14.7%; 95% CI [7.4–27.2] vs. 0.9%; 95% CI [0.3–2.3]; P<0.001) was significantly higher among adolescents who experienced ≥3 adverse psychosocial factors than those who did not experience any such factors. Again, higher prevalence of TU and other SU were also observed among adolescents who reported their parents' use of tobacco or drug (TU: 23.4%; 95% CI [14.5–35.4] vs. 3.3%; 95% CI [2.0–5.4]; P<0.001; and other SU: 4.8%; 95% CI [2.5–8.8] vs. 1.2%; 95% CI [0.5–2.6]; P = 0.001), lack of peer support (TU: 13.4%; 95% CI [7.1–23.9] vs. 8.5%; 95% CI [5.1–13.8]; P = 0.169; and other SU: 6.0%; 95% CI [2.8–12.3] vs. 1.6%; 95% CI [0.8–2.9]; P0.001); and being truant (TU: 15.9%; 95% CI [9.9–24.2] vs. 6.7%; 95% CI [3.3–13.2]; P = 0.029; and other SU: 5.5%; 95% CI [2.6–11.3] vs. 0.8%; 95% CI [0.4–1.6]; P<0.001). The prevalence of TU was higher among adolescents who experienced ≥3 adverse socio-environmental adversities than those who didn't experience any such adversities. Prevalence of tobacco and other substance use among adolescents by gender is displayed in Fig 1. The proportions of using tobacco and other substances were higher among boys than girls (Fig 1).

Table 3 presents the factors associated with tobacco and substance use. Results showed that females were significantly less likely to TU (Adjusted odds ratio [AOR]: 0.21; 95% confidence interval [CI]: 0.13–0.34) and other SU (AOR: 0.42; 95% CI: 0.25–0.70), compared to males.

**Table 2. Prevalence of tobacco and other substance use according to different socio-demographics and adolescent's adverse experiences in Bangladesh: Global School-Based Health Survey (GSHS), 2014.**

| Characteristics | Total (N = 2989) n (%)* | Tobacco use % (95% CI) | P-value ($\chi^2$-test) | Substance use % (95% CI) | P-value ($\chi^2$-test) |
|---|---|---|---|---|---|
| **Total** | | **9.6 (6.3–14.3)** | | **2.3 (1.2–4.3)** | |
| **Age** | | | 0.001 | | 0.265 |
| 11–12 | 104 (2.6) | 3.0 (0.5–16.4) | | 2.9 (0.4–15.9) | |
| 13 | 612 (25.1) | 4.2 (2.2–7.8) | | 3.1 (1.1–8.3) | |
| 14 | 1093 (38.0) | 6.6 (4.5–9.5) | | 2.2 (1.1–4.4) | |
| 15 | 952 (25.9)) | 16.6 (8.7–29.2) | | 0.7 (0.3–2.0) | |
| 16–18 | 221 (8.5) | 16.0 (8.1–29.3) | | 4.1 (1.1–14.4) | |
| **School grade** | | | 0.032 | | 0.580 |
| Class VII | 859 (31.2) | 4.3 (2.1–8.8) | | 2.9 (1.1–7.7) | |
| Class VIII | 328 (25.8) | 5.0 (2.7–9.0) | | 2.6 (1.0–6.7) | |
| Class IX | 1478 (21.8) | 14.9 (9.1–23.6) | | 0.7 (0.3–1.7) | |
| Class X | 305 (21.3) | 16.1 (5.8–37.5) | | 2.3 (0.4–12.9) | |
| **Gender** | | | <0.001 | | 0.013 |
| Male | 1192 (65.3) | 13.4 (8.1–21.3) | | 3.2 (1.5–6.3) | |
| Female | 1788 (34.7) | 1.9 (1.1–3.4) | | 0.5 (0.2–1.5) | |
| **Food insecurity[a]** | | | 0.439 | | 0.654 |
| No | 2488 (85.7) | 9.7 (6.3–14.5) | | 2.2 (1.1–4.1) | |
| Yes | 426 (14.3) | 10.9 (5.2–21.2) | | 1.9 (0.6–5.9) | |
| **Psychosocial factors** | | | | | |
| **Loneliness[b]** | | | 0.009 | | <0.001 |
| No | 2695 (89.1) | 8.4 (4.9–14.1) | | 1.4 (0.6–2.9) | |
| Yes | 281 (10.9) | 16.7 (9.9–26.8) | | 9.1 (4.4–17.4) | |
| **Anxiety-related sleep loss[b]** | | | 0.012 | | <0.001 |
| No | 2840 (95.3) | 8.9 (5.5–13.9) | | 1.6 (0.9–3.0) | |
| Yes | 138 (4.7) | 20.5 (11.7–33.3) | | 14.3 (6.0–30.3) | |
| **Bullied[a]** | | | 0.014 | | 0.001 |
| No | 2367 (75.5) | 6.4 (4.2–9.9) | | 0.9 (0.5–1.8) | |
| Yes | 608 (24.5) | 18.4 (8.6–35.1) | | 6.3 (2.9–12.8) | |
| **No close friends** | | | 0.098 | | 0.899 |
| No | 2701 (91.5) | 9.7 (6.2–14.8) | | 2.3 (1.2–4.3) | |
| Yes | 256 (8.5) | 5.4 (2.3–11.9) | | 2.2 (0.4–9.6) | |
| **Sexual history** | | | <0.001 | | <0.001 |
| No | 2425 (90.1) | 5.7 (2.8–11.3) | | 1.2 (0.6–2.2) | |
| Yes | 218 (9.3) | 47.2 (31.1–64.0) | | 12.6 (4.9–28.7) | |
| **Physically abused[b]** | | | 0.631 | | 0.089 |
| No | 1298 (36.5) | 10.5 (6.0–17.8) | | 1.2 (0.5–2.5) | |
| Yes | 1654 (63.6) | 8.9 (4.9–15.4) | | 2.9 (1.3–5.9) | |
| **No. of adverse psychosocial factors** | | | 0.024 | | 0.001 |
| 0 | 828 (25.6) | 1.1 (0.3–3.7) | | 0.9 (0.3–2.3) | |
| 1 | 1077 (44.3) | 8.8 (4.8–15.6) | | 0.1 (0.03–06) | |
| 2 | 492 (21.6) | 16.4 (5.7–38.8) | | 2.3 (0.9–5.9) | |
| ≥3 | 176 (8.5) | 19.3 (10.9–32.1) | | 14.7 (7.4–27.2) | |
| **Socio-environmental factors** | | | | | |
| **People use tobacco in presence of adolescents** | | | 0.659 | | 0.766 |
| No | 1973 (68.8) | 9.2 (6.2–13.5) | | 2.3 (1.0–4.9) | |
| Yes | 946 (31.2) | 10.5 (5.0–20.6) | | 2.6 (1.3–5.1) | |

(*Continued*)

**Table 2.** (Continued)

| Characteristics | Total (N = 2989) n (%)* | Tobacco use % (95% CI) | P-value (χ²-test) | Substance use % (95% CI) | P-value (χ²-test) |
|---|---|---|---|---|---|
| **Parental tobacco or drug use** | | | <0.001 | | 0.001 |
| No | 2177 (70.3) | 3.3 (2.0–5.4) | | 1.2 (0.5–2.6) | |
| Yes | 784 (29.7) | 23.4 (14.5–35.4) | | 4.8 (2.5–8.8) | |
| **Poor understanding with parents[a]** | | | 0.003 | | 0.788 |
| No | 1430 (47.6) | 4.5 (2.4–8.5) | | 2.3 (1.1–4.6) | |
| Yes | 1441 (52.4) | 14.1 (8.6–22.4) | | 2.2 (1.1–4.3) | |
| **Poor parental monitoring[a]** | | | 0.003 | | 0.524 |
| No | 1342 (43.4) | 4.5 (2.6–7.7) | | 2.0 (0.8–4.6) | |
| Yes | 1521 (56.7) | 13.3 (7.9–21.5) | | 2.4 (1.3–4.6) | |
| **Lack of peer support[a]** | | | 0.169 | | 0.001 |
| No | 1784 (56.0) | 8.5 (5.1–13.8) | | 1.6 (0.8–2.9) | |
| Yes | 1062 (44.0) | 13.4 (7.1–23.9) | | 6.0 (2.8–12.3) | |
| **Truancy[a]** | | | 0.029 | | <0.001 |
| No | 1856 (69.1) | 6.7 (3.3–13.2) | | 0.8 (0.4–1.6) | |
| Yes | 1026 (30.9) | 15.9 (9.9–24.2) | | 5.5 (2.6–11.3) | |
| **No. of adverse socio-environmental factors** | | | <0.001 | | 0.001 |
| 0 | 356 (10.2) | 0.7 (0.1–2.9) | | 1.4 (0.2–6.2) | |
| 1 | 609 (21.7) | 2.9 (1.3–6.6) | | 1.4 (0.7–2.7) | |
| 2 | 662 (28.2) | 4.7 (2.4–9.2) | | 0.5 (0.1–2.3) | |
| ≥3 | 1097 (39.8) | 18.9 (11.2–30.1) | | 4.4 (2.2–8.8) | |

'Note:

*Numbers are unweighted and percentages are weighted. Percentages may not total 100.0 because of rounding.

[a]In the past 12 months

[b]in the past 30 days.

The likelihood of TU and other SU was higher among adolescents who reported being bullied (TU: AOR: 1.93; 95% CI: 1.24–3.00; and other SU: AOR: 3.43; 95% CI: 1.46–7.99) and having sexual history (TU: AOR: 19.38; 95% CI: 12.43–30.21; and other SU: AOR: 5.34; 95% CI: 2.17–13.29). Moreover, anxiety-related sleep loss was significantly associated with adolescents' TU behavior (AOR: 2.41; 95% CI: 1.02–5.82) whereas the likelihood of other SU (AOR: 3.27; 95% CI: 1.14–9.44) was higher among adolescents reporting loneliness. Among adverse socio-environmental factors, parental tobacco or drug use (AOR: 7.81; 95% CI: 5.08–12.01), poor understanding with parents (AOR: 2.22; 95% CI: 1.36–3.65), poor parental monitoring (AOR: 1.96; 95% CI: 1.16–3.31), and lack of peer support (AOR: 3.13; 95% CI: 1.84–5.31) were significantly associated with higher likelihood of adolescents' tobacco use. However, adolescents with lack of peer support (AOR: 2.45; 95% CI: 1.02–5.91) and higher truancy (AOR: 4.29; 95% CI: 1.81–10.12) were more likely to use other substances (Table 3).

Experience of increased number of adverse psychosocial adversities was found to be associated with greater tobacco use in a graded manner (Table 4). After adjusting for age, gender, grade and food insecurity, higher odds of tobacco use were observed among adolescents who reported 1 (AOR: 4.36 times; 95% CI: 1.34–14.24), 2 (AOR: 8.69 95% CI: 1.67–28.23), and ≥3 (AOR: 17.46; 95% CI: 6.20–49.23) adverse psychosocial experiences than who did not report any psychosocial events. Additionally, adolescents who experienced 2 and ≥3 adverse socio-environmental events were 5.94 times and 27.85 times, respectively, more likely to report using tobacco. The likelihood of other substance use was found to be higher among adolescents who experienced ≥3 (AOR: 13.71, 95% CI: 3.19–58.93) adverse psychosocial events.

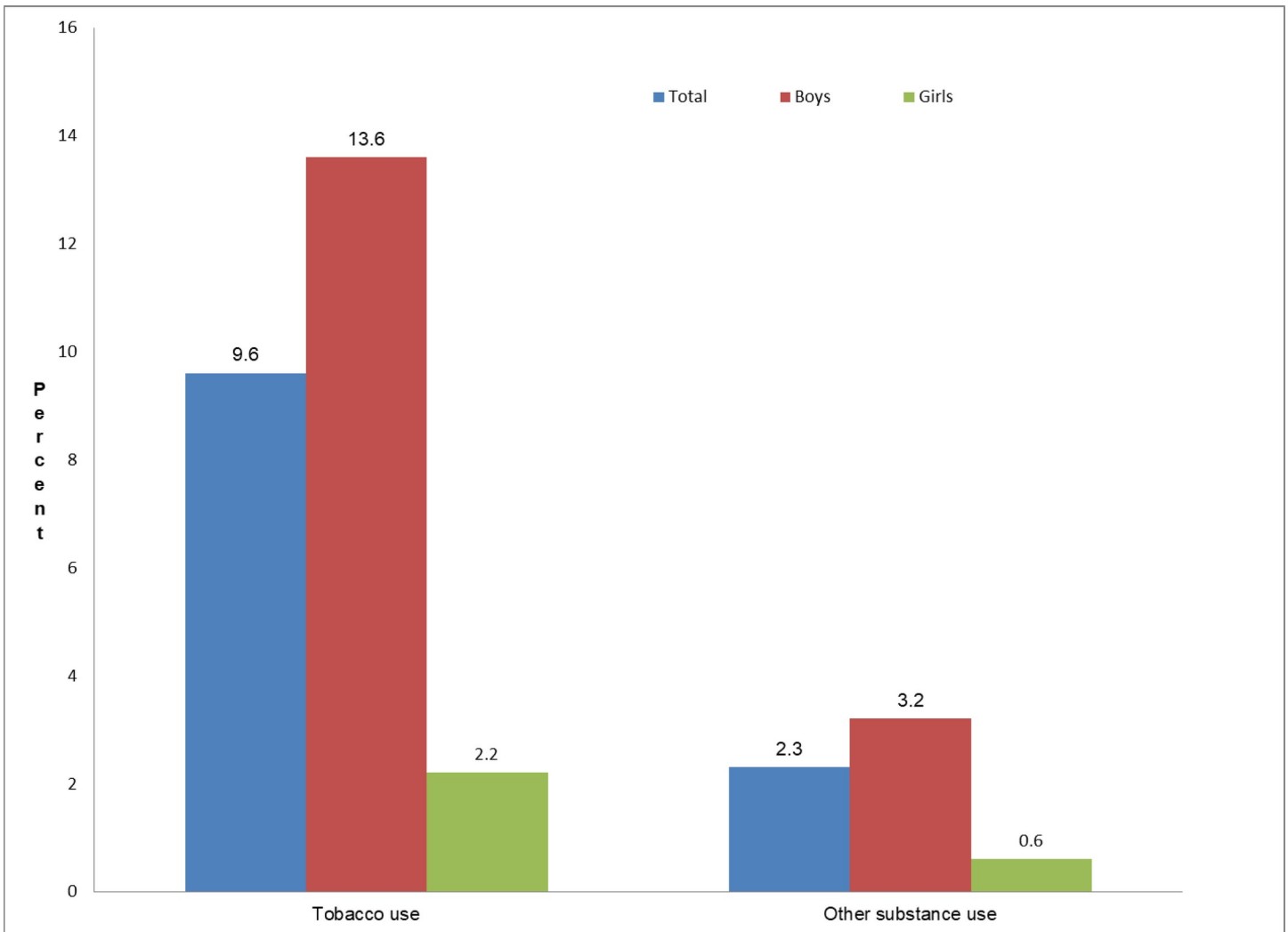

**Fig 1. Prevalence of tobacco and other substance use among adolescents by gender, the 2014 Global School-based Student Health Survey (GSHS), Bangladesh.**

However, multiple adverse socio-environmental experience was not found to be associated with adolescents' other substance use.

## Discussion

In this study, using a nationally representative data, we estimate the prevalence and factors associated, psychosocial and socio-environmental factors particularly, with tobacco and other illegal substances use among school-going adolescents in Bangladesh. Our findings showed that 9.6% adolescents use any form of tobacco (smoke cigarette or use other tobacco products), and 2.3% use any other substance (alcohol and/or marijuana). The prevalence of tobacco use found in this study is consistent with the findings from earlier studies in Bangladesh [5, 28–30]. Although there is no study that estimated the prevalence of other substance use in Bangladesh, the prevalence reported in this study is much lower than that of other Asian [25, 41] and African countries [42, 43]. This difference could be due to underreporting that could be explained by two factors: strong social stigma and legal prohibition of alcohol use and other illicit drug use in Bangladesh [44]. Moreover, Bangladesh is a predominantly Muslim country,

**Table 3. Results of the logistic regression analyses of the relationship between socio-demographics and adolescent's adverse experiences, and tobacco and other substance use in Bangladesh: Global School-Based Health Survey (GSHS), 2014.**

| Risk factors | Tobacco use | Other substance use |
|---|---|---|
| | AOR (95% CI)[a] | AOR (95% CI)[a] |
| **Age** | | |
| 11–12 years[(RC)] | 1.00 | 1.00 |
| 13 years | 6.03 (1.49–25.36) * | 2.91 (0.59–14.36) |
| 14 years | 2.91 (0.63–12.59) | 1.04 (0.18–5.87) |
| 15 years | 3.82 (0.79–18.50) | 0.97 (0.15–6.57) |
| 16–18 years | 3.25 (0.64–16.49) | 1.81 (0.26–12.76) |
| **School grade** | | |
| Class 7[(RC)] | 1.00 | 1.00 |
| Class 8 | 1.72 (0.59–4.99) | 1.15 (0.41–3.28) |
| Class 9 | 3.09 (1.17–8.12) * | 0.34 (0.13–0.93) |
| Class 10 | 5.00 (1.75–14.28) ** | 0.73 (0.17–3.14) |
| **Gender** | | |
| Male[(RC)] | 1.00 | 1.00 |
| Female | 0.21 (0.13–0.34) *** | 0.42 (0.25–0.70) *** |
| **Food insecurity** | | |
| No[(RC)] | 1.00 | 1.00 |
| Yes | 0.72 (0.43–1.21) | 0.92 (0.33–2.55) |
| **Adverse Psychosocial Factors** | | |
| **Loneliness** | | |
| No[(RC)] | 1.00 | 1.00 |
| Yes | 1.51 (0.77–2.96) | 3.27 (1.14–9.44) * |
| **Anxiety** | | |
| No[(RC)] | 1.00 | 1.00 |
| Yes | 2.41 (1.02–5.82) * | 2.61 (0.77–8.85) |
| **Bullied** | | |
| No[(RC)] | 1.00 | 1.00 |
| Yes | 1.93 (1.24–3.00) ** | 3.43 (1.46–7.99) ** |
| **No close friends** | | |
| No[(RC)] | 1.00 | 1.00 |
| Yes | 0.47 (0.18–1.22) | 0.34 (0.05–2.11) |
| **Sexual history** | | |
| No[(RC)] | 1.00 | 1.00 |
| Yes | 19.38 (12.43–30.21) *** | 5.34 (2.17–13.29) *** |
| **Physically abused** | | |
| No[(RC)] | 1.00 | 1.00 |
| Yes | 0.80 (0.52–1.23) | 1.76 (0.63–4.87) |
| **Adverse Socio-Environmental Factors** | | |
| **People use tobacco in presence of adolescents** | | |
| No[(RC)] | 1.00 | 1.00 |
| Yes | 0.88 (0.59–1.28) | 1.28 (0.59–2.79) |
| **Parental tobacco or drug use** | | |
| No[(RC)] | 1.00 | 1.00 |
| Yes | 7.81 (5.08–12.01) *** | 1.62 (0.72–3.63) |
| **Poor understanding with parents** | | |
| No[(RC)] | 1.00 | 1.00 |

(*Continued*)

**Table 3.** (Continued)

| Risk factors | Tobacco use | Other substance use |
|---|---|---|
| | AOR (95% CI)[a] | AOR (95% CI)[a] |
| Yes | 2.22 (1.36–3.65) ** | 1.05 (0.43–2.53) |
| **Poor parental monitoring** | | |
| No[(RC)] | 1.00 | 1.00 |
| Yes | 1.96 (1.16–3.31) ** | 0.81 (0.35–1.88) |
| **Lack of peer support** | | |
| No[(RC)] | 1.00 | 1.00 |
| Yes | 3.13 (1.84–5.31) *** | 2.45 (1.02–5.91) * |
| **Truancy** | | |
| No[(RC)] | 1.00 | 1.00 |
| Yes | 1.08 (0.70–1.65) | 4.29 (1.81–10.12) ** |

**Note:**

[a]Model was adjusted for all the variables included in this table. Values with superscript asterisks *, **, and *** indicate $p < 0.05$, $p < 0.01$, and $p < 0.001$, respectively. (RC): Reference category, AOR: adjusted odds ratio, CI: confidence interval.

and Islam provides clear ruling against using any intoxicants [45]. Nonetheless, the findings of this study represent a matter of great concern because adolescent tobacco use control is critical to the control of overall tobacco use in the country. Moreover, starting tobacco use at adolescence provides some insights into the potential developmental sequence of various drug dependence. For example, smoking or drinking alcohol precedes the use of marijuana, which in turn precedes the use of other illicit substances, such as cocaine or heroin [46]. Therefore, these findings highlight the importance of early prevention and intervention strategies.

This study identified several socio-demographic, psychosocial, and socio-environmental risk factors that are associated with adolescents' tobacco and other substance use in

**Table 4. Association between multiple adverse experiences and adolescents' tobacco and other substance use in Bangladesh, 2014.**

| Multiple Adverse Experiences | Tobacco use | Other substance use |
|---|---|---|
| | AOR (95% CI) [a] | AOR (95% CI) [a] |
| **No. of adverse psychosocial factors** | | |
| 0[(RC)] | 1.00 | 1.00 |
| 1 | 4.36 (1.34–14.24) ** | 0.13 (0.18–1.00) |
| 2 | 8.69 (1.67–28.23) *** | 2.26 (0.54–9.44) |
| ≥3 | 17.46 (6.20–49.23) *** | 13.71 (3.19–58.93) ** |
| **No. of adverse socio-environmental factors** | | |
| 0[(RC)] | 1.00 | 1.00 |
| 1 | 4.42 (0.77–25.38) | 0.92 (0.14–6.05) |
| 2 | 5.94 (1.14–31.06) * | 0.08 (0.01–1.00) |
| ≥3 | 27.85 (5.77–134.52) *** | 3.34 (0.49–22.73) |

**Note:**

[a]Model was adjusted for age, gender, school grade and food insecurity. Values with superscript asterisks *, **, and *** indicate $p < 0.05$, $p < 0.01$, and $p < 0.001$, respectively. (RC): Reference category, AOR: adjusted odds ratio, CI: confidence interval.

Bangladesh. Consistent to a prior study in Bangladesh [30], and other studies across the world [47, 48], this study found that male adolescents are more likely to use tobacco or other substances than female adolescents. One of the reasons could be gender norms and that male adolescents are less monitored or supervised than females. Earlier studies documented that boys are generally less monitored by parents than girls [49], and low parental monitoring and supervision are associated with higher tobacco and alcohol use among adolescents [15, 20]. However, some studies in Bangladeshi did not find any significant relationship between gender and tobacco use in adolescence [28, 29]. More research in gender differences in tobacco use is needed to clarify the inconsistence in findings.

Consistent with earlier studies [25, 50], this study observed an increased likelihood of substance use (alcohol or marijuana) among lonely adolescents. Adolescents who are isolated from peers may exhibit greater anti-social behaviors and deviance than socially connected adolescents [51], which, in turn, can manifest in increased substance use [52]. We also found that anxiety-related sleep loss was associated with adolescents' higher tobacco use, which is supported by earlier studies [53, 54]. Perhaps, adolescents with persistent sleep loss due to anxiety or stress may smoke or use other tobacco products to self-medicate with mild stimulant effect of nicotine to improve their mood [55].

Bullying victimization was found to be significantly associated with both tobacco use and other substance use in this study. The possible reasons could be bullying victimization may cause severe anxiety and depression [56]. To reduce the anxiety or to increase their social image among their peers, victims may use tobacco or other substances [57]. Victimized adolescents may also use tobacco, alcohol, or marijuana to alleviate the isolation, helplessness, and low self-esteem associated with victimization [58]. Sexual history was also found to be strongly associated with tobacco and other substance use among adolescents. Engaging in sexual activity in early adolescence has been seen as a problem behavior, because those adolescents often face challenges in many areas of life [59]. Particularly, problem arises when sexual activity took place prematurely. Certain emotional maturity might be lacking among young adolescents that increases their risks of engaging in unsafe sexual contacts [60]. Sexual activity in these periods can make adolescents vulnerable to developing mental disorder like depression or anxiety [61, 62], which, in turn, could lead them to use tobacco or other substances [63, 64].

We found that several socio-environmental factors are likely to be associated with adolescents' unhealthy behaviors. For instance, we found that parental tobacco or drug use is a risk factor for offspring tobacco and other substance use, which is in parallel with prior studies across the world [33, 65–67]. Possible explanations could be normalization of smoking in the home environment and easy access to these substances at home [66, 67]. We also found that truancy is positively associated with adolescents' substance use. Substance use is expected to be higher among truant adolescents than their counterparts because truant adolescents may have more unsupervised time, less parental monitoring, and spend more time with deviant peers [68], all of which could lead to problem behaviors.

We found adolescents' experience of adverse events, both psychosocial and socio-environmental, to be associated with tobacco use in a graded manner. For example, the experience of one or more psychosocial events was associated with a stepwise elevation in the likelihood of adolescents' tobacco use. This finding corroborates with a recent study which documented that every additional psychosocial risk factor was associated with an estimated 100% increase in the odds using tobacco [69]. These finding highlights the importance of considering the co-occurring risk factors associated with tobacco use. However, we did not find statistically significant association between experience of multiple adverse socio-environmental events and adolescents' other substance use, which could be due to very few positive responses in the multiple adverse socio-environmental categories.

This study has several strengths. Firstly, we used a nationally representative large sample with appropriate statistical methods by taking into account complex survey design and sample weights. Secondly, the GSHS uses same standardized methods regarding the sample (school-based), data collection, and wording of questions across surveys, which enables international comparisons of the results. However, this study is not without limitations. First, because GSHS is a cross-sectional survey, directionality of the assessed relationship cannot be ascertained. Second, since adolescents were asked to report tobacco, alcohol and marijuana use within the past 30 days, the results are subject to recall bias. In addition, Bangladesh is a Muslim country where alcohol consumption and marijuana use are illegal; therefore, an underestimate of the prevalence of these substances is possible due to social desirability bias [70]. Moreover, adolescent tobacco, alcohol or marijuana use may be underreported, because the GSHS was restricted to adolescents who attended school and were present on the day of the survey. Third, as we used secondary data, analyses were limited to the variables that were available. We were unable to include other relevant variables such as, availability, accessibility, and price of substances, etc. Finally, since the sample only included school-going adolescents, findings may not be generalizable to out-of-school adolescents.

Limitations notwithstanding, this study has the potential to inform the development and/or modification of prevention strategies. Teachers, parents or other school personnel could play important role in early identification of adolescents with these emotional and behavioral problems. However, they would require appropriate training (e.g. mental health training) and professional development opportunities to build their skills and self-efficacy to be able to recognize, and appropriately respond to the adolescents with these problems. Furthermore, the findings suggest the need for incorporating mental health education programs in school-curricula; and the importance of the family and social connections among these populations [71]. Findings of this study underscore the importance of parental awareness to help and support adolescents in problems. Finally, since parental smoking increases the likelihood of adolescent smoking, universal prevention approaches which combine tobacco cessation campaign through mass media, including school- and community-based programs could be effective [72].

## Conclusions

This study, using a nationally representative data, estimates the prevalence and risk factors associated with tobacco and other substance use among school-going adolescents in Bangladesh. The findings of this study have important implications for public health professionals and practitioners who work on adolescent substance use programs. Several adverse psychosocial (loneliness, anxiety, bullying, and sexual history), and socio-environmental factors (parental tobacco use, lack of peer support, and poor parental monitoring) were found to be significantly associated with adolescents' tobacco and other substance use. Although the relationships are naturally complex, some patterns are discernible. Further studies with longitudinal data are needed to determine the mechanisms, processes and directionality of these relationships.

## Acknowledgments

The authors are thankful to the Department of Chronic Diseases and Health Promotion, World Health Organization, jointly worked with Center for Disease Control and Prevention (CDC), for permitting us to use the datasets for this analysis. We are also grateful to the Department of Population Science and Human Resource Development, University of Rajshahi, Bangladesh where the study was conducted.

## Author Contributions

**Conceptualization:** Md. Mostaured Ali Khan, Md. Golam Mustagir.

**Data curation:** Md. Mostaured Ali Khan, Md. Golam Mustagir, Md. Sharif Kaikobad.

**Formal analysis:** Md. Mostaured Ali Khan, Md. Mosfequr Rahman, Md. Rajwanul Haque.

**Methodology:** Md. Mostaured Ali Khan, Md. Mosfequr Rahman, Md. Rajwanul Haque.

**Software:** Md. Mostaured Ali Khan, Md. Golam Mustagir, Md. Sharif Kaikobad.

**Supervision:** Md. Mosfequr Rahman, Syeda S. Jeamin.

**Writing – original draft:** Md. Mostaured Ali Khan, Md. Mosfequr Rahman, Md. Rajwanul Haque.

**Writing – review & editing:** Md. Mosfequr Rahman, Syeda S. Jeamin.

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
