## [Decision Letter · Decision Letter 0]

13 Aug 2020

PONE-D-20-18742

Psychosocial and socio-environmental factors associated with adolescents’ tobacco and substance use in Bangladesh

PLOS ONE

Dear Dr. Rahman,

Thank you for submitting your manuscript to PLOS ONE. After careful consideration, we feel that it has merit but does not fully meet PLOS ONE’s publication criteria as it currently stands. Therefore, we invite you to submit a revised version of the manuscript that addresses the points raised during the review process.

We look forward to receiving your revised manuscript.

Kind regards,

Stanton A. Glantz

Academic Editor

PLOS ONE

Journal Requirements:

2.We suggest you thoroughly copyedit your manuscript for language usage, spelling, and grammar. If you do not know anyone who can help you do this, you may wish to consider employing a professional scientific editing service.  

3.Thank you for stating the following financial disclosure:

 [The funders had no role in study design, data collection and analysis, decision to publish, or preparation of the manuscript.].

Reviewers' comments:

Reviewer's Responses to Questions

**Comments to the Author**

1. Is the manuscript technically sound, and do the data support the conclusions?

Reviewer #1: Yes

Reviewer #2: Partly

2. Has the statistical analysis been performed appropriately and rigorously? 

Reviewer #1: Yes

Reviewer #2: I Don't Know

3. Have the authors made all data underlying the findings in their manuscript fully available?

Reviewer #1: No

Reviewer #2: No

4. Is the manuscript presented in an intelligible fashion and written in standard English?

Reviewer #1: Yes

Reviewer #2: Yes

5. Review Comments to the Author

Reviewer #1: This is a study using on the psychosocial and socio-environmental factors associated with tobacco use and substance use (combined alcohol and/or marijuana use). They find a dose-response relationship between adverse psychosocial factors and tobacco use. The likelihood of substance use (alcohol and/or marijuana) was higher among adolescents who experienced 3+ adverse psychosocial events. Tobacco and substance use in an adjusted logistic analysis, those who were bullied, and had ‘adverse sexual history’ whatever that means. Substance use is associated was associated with loneliness, lack of peer support and higher. Tobacco use was associated with anxiety-related sleep loss parental tobacco or drug use, poor parental understanding, poor parental monitoring and lack of peer support.

This study is an important update of data from 2014 to the previously published papers on adolescent tobacco and substance use using the 2007 data from the Global School-based Student Health Survey for Bangladesh.

1. You must say where the publicly available database is located and how to access it

2. CDC is Centers for Disease Control and Prevention

3. Explain what you mean by adverse sexual history. If it means yes = have had sex, no = have not had sex, then you should call the variable sexual history or something less loaded than adverse sexual history. I see that you do that in Table 1. Call the variable sexual history or some other less loaded term throughout the text

4. Line 247: male adolescents are less monitored or supervised.

5. Table 3: Physical abused should be physically abused

6. Explain why you decided to lump together alcohol and marijuana use into substance use. For a general reader from a non-Muslim majority country, this may seem strange

7. Table 2: the subscripts that you give to the table – a) in past 12 months, b) in past 30 days – are never used in Table 2.

Reviewer #2: Psychosocial and socio-environmental factors associated with adolescents’ tobacco

and substance use in Bangladesh

Reviewer comments to authors

Overall, this is an important study that examines adolescent substance use and tobacco use, and underlying factors of such use among adolescents in Bangladesh. The manuscript would benefit from a more in-depth description of the problem and context. The manuscript is lacking several aspects of the literature, importantly, the gaps that the study data fill and how the approach in this study will help future research, policy and practice. Since the data are from 2014, the authors may need to justify why they sought to study prevalence when more recent data have already been published from 2017-18. Several aspects of the methods could include greater detail and justification. The study would also do well to include more details in the Discussion based on comparisons with other low-and middle-income countries or other similar socio-religious contexts. Specifically, the authors should modify their conclusion that this study provides information about predictors of tobacco and substance use as it is a cross-sectional study and not a longitudinal one.

Abstract

The authors may want to be consistent in saying use/ misuse/ abuse.

At the outset in the abstract, it is unclear as to how the authors’ have used a cross-sectional study design to ascertain predictors of tobacco and substance use. The description of results indicates that the authors have assessed associations/ relationships between such use (TU and SU) and factors. In multiple places the authors have stated that they determined predictors, which is not feasible using this study design.

Provide Odds Ratios for: “Experience of adverse socio-environmental factors, such as parental substance use, poor monitoring and understanding, and lack of peer support were also positively

associated with TU and/or SU.”

It is not clear what the authors mean by: “Additionally, multiple adverse psychosocial factors were associated with TU in a graded manner.”

It is not clear what the authors mean by: “Tobacco and substance use among school-going adolescents in Bangladesh are relatively prevalent.” Relative to what percentage, what age group, and where?

Why was “substance use” created as an outcome measure? Why did the authors not use any other validated measure on substance use? What forms of marijuana use were included?, since ‘drinking’ alcohol is specified, it would be good to maintain parallelism in the sentence.

Introduction

If it is known that 6.9% of adolescents are tobacco users, and 2% are alcohol users from 2017-2018 data, which is based on more recent data than the study under review, what is the justification for this study based on data from 2014? What is the gap that the current study fills in terms of assessing prevalence?

Suggestion to include a description of the types of tobacco use in Bangladesh, as there may be many diverse forms of use. Line 99 – “limited to only tobacco (smoke or smokeless)”

What makes the authors believe that the reasons for adolescents using tobacco in global studies differ from the reasons why adolescents use in Bangladesh?

Line 98-99 What is the reason why the authors want to include an urban sample on tobacco users? What proportion of Bangladesh is urban?

Line 100: What is the evidence that psychosocial factors are important? Perhaps the authors would like to define these factors, explain briefly how they are likely interlinked. In the paragraph on global studies (line 78-89) the authors mention many factors, but latel them interpersonal risk factors, protective factors, psychosocial distress, and so one is not sure what psychosocial factors entail exactly?

Line 106: MAE – Multiple adverse experiences – This construct is mentioned without any citation or background information. It seems like it is better suited as a measure rather than mentioned in passing at the end of the introduction.

Data and Methods

Line 119: What is the disadvantage of using a school-based sample? What is the proportion of adolescents in Bangladesh who do not go to school and are using tobacco?

How long was the self-administered questionnaire?

What steps were taken by the study team to manage adolescent distress while answering questions on Multiple adverse experiences?

Was student assent collected?

What language was the survey completed in and was it in the local language? Do participating adolescents understand and use the term marijuana colloquially?

In the creation of age-standardized weights, what specific marginal population proportions were used from the population and housing census, and why? Please provide this information in a supplemental table. Or is it just based on age?

See Line 160-164: Multiple imputation will not likely make the sample more representative. The citation does not match the text preceding it and it is indistinct why multiple imputation was used. It would also be useful to explain why “ever had sexual intercourse” was included in this context: “To ensure representativeness and to prevent misinterpretation or any form of biases in the analysis, the allocation of missing values was done through a multiple imputation method using a logistic regression model by taking into account the known values, most frequently for “ever had sexual intercourse” [34].”

The measures include no mention of MAE.

On what basis were the measures selected? And why are they appropriate in Bangladesh? The adverse events description and combination of measures in Table 1 appear a bit arbitrary and not evidence-based. Suggestion to add a justification and description of how these measures were categorized and used based on the literature.

Results

Line 175-176: Please clarify what is meant by “more than half of the adolescents had poor parental understanding?”

Line 179-189: Are these differences significant, if so please provide summary of test statistics (Confidence Intervals and p-values)?

Line 185 - factors than (those) who did not experience any (such factors).

Figure 1 does not show that boys are significantly more likely to use tobacco and substances. It only shows that the proportion of boys using tobacco is higher than girls, and only marginally so for substances.

How are the authors defining an adverse sexual history? The measures in Table 1 do not include this – it just asks, ‘Have you ever had sexual intercourse?’

Line 209: The use of dose-response relationship appears inappropriate in this context as there is no experiment underway. The analysis shows an additive negative effect of adverse experiences.

Line 210-211: Why were both age and grade adjusted for as confounders?

Tables 2 should provide sample N and counts in a column on unweighted data.

Line 220: This seems like a sentence more appropriate as an explanation in the Discussion section.

Discussion

Line 227-229: Reference 26-28 are much more dated than prevalence rates reported in the Introductory paragraph.

Line 233: remove extra space before period

Line 267-270: It may be useful to bring this point earlier in the paper, maybe as a footnote to Table 1.

Line 275- predictors is not an accurate description of this study’s findings

Line 280-289 and line 242-252: The idea of parental involvement/ monitoring and understanding is repetitive and may be combined effectively.

Study strengths and implications should be described in greater detail.

Please review: https://www.childwelfare.gov/topics/preventing/preventionmonth/resources/ace/

For other descriptions of adverse childhood events and categorization.

Line 314 – The authors themselves provide other prevalence estimates from Bangladesh, so it may seem prudent to avoid saying it is the first or even most recent?

It may be useful to discuss mental health promotion opportunities in light of psychosocial findings.

6. PLOS authors have the option to publish the peer review history of their article (what does this mean?). If published, this will include your full peer review and any attached files.

Reviewer #1: No

Reviewer #2: No

---

## [Author Response · Author response to Decision Letter 0]

7 Sep 2020

September 8, 2020

Prof. Stanton A. Glantz

Academic Editor

PLOS ONE

Re: Manuscript Number PONE-D-20-18742

Dear Prof. Stanton A. Glantz,

Thank you very much for your kind consideration of our manuscript entitled, “Psychosocial and socio-environmental factors associated with adolescents’ tobacco and substance use in Bangladesh”. We have revised the manuscript according to the journal’s requirements, editor’s comments and to the comments by the reviewers. Their useful and productive comments helped us to improve the clarity and quality of the manuscript. Where we have changed the text, the corresponding sentences in the text have been highlighted with the track changes function. The page and line numbers referred to are those in the margins of our clean revised manuscript. Where textual changes are included in this response to reviewer, we have indented and italicized them, and removed references.

It is important to mention here that this study did not receive grant from any funding agency. Unfortunately, we provided the following statement while submitting the manuscript initially: The funders had no role in study design, data collection and analysis, decision to publish, or preparation of the manuscript. Since we did not receive any grant please fill-up the appropriate option for that.

We hope that the revisions are satisfactory in addressing the issues raised by the editor and the reviewers, and look forward to hearing your decision about this article.

Yours sincerely,

Md. Mosfequr Rahman, PhD

Department of Population Science and Human Resource Development

University of Rajshahi

Rajshahi-6205, Bangladesh.

E-mail: mosfeque@ru.ac.bd

Journal Requirements:

 Response: Thank you. We have revised the manuscript according the guidelines of PLOS ONE.

2.We suggest you thoroughly copyedit your manuscript for language usage, spelling, and grammar. If you do not know anyone who can help you do this, you may wish to consider employing a professional scientific editing service. 

Response: Thank you. This manuscript has been copyedited thoroughly for language usage, spelling, and grammar by one of our co-author Syeda S. Jesmin, who is an Associate Professor of Sociology and Psychology, University of North Texas at Dallas, Dallas, TX, USA. She can be contacted by the following e-mail: syeda.Jesmin@untdallas.edu.

3.Thank you for stating the following financial disclosure:

 [The funders had no role in study design, data collection and analysis, decision to publish, or preparation of the manuscript.].

a. Please clarify the sources of funding (financial or material support) for your study. List the grants or organizations that supported your study, including funding received from your institution.

d. If you did not receive any funding for this study, please state: “The authors received no specific funding for this work.”

Response: The authors received no specific funding for this work.

Reviewer #1

This is a study using on the psychosocial and socio-environmental factors associated with tobacco use and substance use (combined alcohol and/or marijuana use). They find a dose-response relationship between adverse psychosocial factors and tobacco use. The likelihood of substance use (alcohol and/or marijuana) was higher among adolescents who experienced 3+ adverse psychosocial events. Tobacco and substance use in an adjusted logistic analysis, those who were bullied, and had ‘adverse sexual history’ whatever that means. Substance use is associated was associated with loneliness, lack of peer support and higher. Tobacco use was associated with anxiety-related sleep loss parental tobacco or drug use, poor parental understanding, poor parental monitoring and lack of peer support.

This study is an important update of data from 2014 to the previously published papers on adolescent tobacco and substance use using the 2007 data from the Global School-based Student Health Survey for Bangladesh.

Response: Thank you for your appreciations. 

1. You must say where the publicly available database is located and how to access it

Response: Thank you very much. We have provided the following links in the text for the location of GSHS data (Page 7; Lines: 143-144). 

A detailed description of the GSHS and data used in this study are available at: https://extranet.who.int/ncdsmicrodata/index.php/catalog/485 and https://www.cdc.gov/gshs/countries/seasian/bangladesh.htm

2. CDC is Centers for Disease Control and Prevention

Response: Yes, it is. We have corrected it in the manuscript.

3. Explain what you mean by adverse sexual history. If it means yes = have had sex, no = have not had sex, then you should call the variable sexual history or something less loaded than adverse sexual history. I see that you do that in Table 1. Call the variable sexual history or some other less loaded term throughout the text

Response: Thank you. We called this variable as “sexual history” and corrected accordingly throughout the text and tables.

4. Line 247: male adolescents are less monitored or supervised.

Response: We have revised it accordingly.

5. Table 3: Physical abused should be physically abused 

Response: We have corrected it accordingly.

6. Explain why you decided to lump together alcohol and marijuana use into substance use. For a general reader from a non-Muslim majority country, this may seem strange

Response: Thank you very much. We have added alcohol and marijuana use together and named the variables as “substance use”, because both of these substances are illegal in Bangladesh. Smoking cigarette is not illegal but against the cultural norms and also unacceptable. We wanted to differentiate these two. The following sentence is added to the text for clarifying it (Page 8, Lines 153-154):

“Substance use (SU) was assessed combining the use of two illegal substances- alcohol and marijuana”

7. Table 2: the subscripts that you give to the table – a) in past 12 months, b) in past 30 days – are never used in Table 2.

Response: Thank you. We are sorry for such unintentional mistakes. We have corrected it in Table 2.

Reviewer #2

Overall, this is an important study that examines adolescent substance use and tobacco use, and underlying factors of such use among adolescents in Bangladesh. The manuscript would benefit from a more in-depth description of the problem and context. The manuscript is lacking several aspects of the literature, importantly, the gaps that the study data fill and how the approach in this study will help future research, policy and practice. Since the data are from 2014, the authors may need to justify why they sought to study prevalence when more recent data have already been published from 2017-18. Several aspects of the methods could include greater detail and justification. The study would also do well to include more details in the Discussion based on comparisons with other low-and middle-income countries or other similar socio-religious contexts. Specifically, the authors should modify their conclusion that this study provides information about predictors of tobacco and substance use as it is a cross-sectional study and not a longitudinal one.

Response: Thank you very much. We have addressed the issues one by one in the following sections.

Abstract

The authors may want to be consistent in saying use/ misuse/ abuse.

Response: Thank you very much. We have used ‘use’ consistently in the abstract and text, and corrected accordingly.

At the outset in the abstract, it is unclear as to how the authors’ have used a cross-sectional study design to ascertain predictors of tobacco and substance use. The description of results indicates that the authors have assessed associations/ relationships between such use (TU and SU) and factors. In multiple places the authors have stated that they determined predictors, which is not feasible using this study design.

Response: Thank you. Yes, predictors should not be used in cross-sectional study design. We have corrected it throughout the manuscript where we use predictors.

Provide Odds Ratios for: “Experience of adverse socio-environmental factors, such as parental substance use, poor monitoring and understanding, and lack of peer support were also positively associated with TU and/or SU.”

Response: We have provided odds ratios in abstract in our revised manuscript.

It is not clear what the authors mean by: “Additionally, multiple adverse psychosocial factors were associated with TU in a graded manner.”

Response: While we examined the relationship between adolescents’ multiple adverse experiences with TU and SU, we observed a graded relationship between psychosocial factors and TU. This means that the odds of TU are gradually increasing with the increment of multiple experiences of adverse psychosocial events. We have corrected the sentence as follows for clear understanding (Page 3, Lines 45-46).

“Additionally, adolescents’ multiple adverse experience of psychosocial factors was found to be associated with TU in a graded manner.” 

It is not clear what the authors mean by: “Tobacco and substance use among school-going adolescents in Bangladesh are relatively prevalent.” Relative to what percentage, what age group, and where?

Response: Thank you. The prevalence TU and SU that we found in our study was much lower than the prevalence found in some other developing countries like Ethiopia, Ghana, Philippines, Namibia, Chili etc (References given below). These studies were also used the data of respective country’s GSHS. Therefore, we wrote relatively. Since it creates confusion, we deleted the term ‘relatively’ from the sentence. 

Page, R.M., et al., Psychosocial distress and substance use among adolescents in four countries: Philippines, China, Chile, and Namibia. Youth & society, 2011. 43(3): p. 900-930.

Pengpid, S. and K. Peltzer, Alcohol use and misuse among school-going adolescents in Thailand: results of a national survey in 2015. International journal of environmental research and public health, 2019. 16(11): p. 1898

Birhanu, A.M., T.A. Bisetegn, and S.M. Woldeyohannes, High prevalence of substance use and associated factors among high school adolescents in Woreta Town, Northwest Ethiopia: multi-domain factor analysis. BMC public health, 2014. 14(1): p. 1186.

Hormenu, T., et al., Psychosocial Determinants of Marijuana Utilization Among Selected Junior High School Students in the Central Region of Ghana. Journal of Preventive Medicine and Care, 2018. 2(2): p. 43.

Why was “substance use” created as an outcome measure? Why did the authors not use any other validated measure on substance use? 

Response: Thank you. Alcohol, marijuana, and tobacco are substances most commonly used by adolescents (References below).

Johnson LD, O’Malley PM, Bachman JG, Schulenberg JE, Miech RA. Monitoring the Future national survey results on drug use, 1975-2013: Volume 1, Secondary school students. Ann Arbor, MI: Institute for Social Research, University of Michigan, 2014: 32-36.

Latimer, W. and J. Zur, Epidemiologic trends of adolescent use of alcohol, tobacco, and other drugs. Child and Adolescent Psychiatric Clinics, 2010. 19(3): p. 451-464.

World Drug Report 2018 (United Nations publication, Sales No. E.18.XI.9)

Ramo, D.E., H. Liu, and J.J. Prochaska, Tobacco and marijuana use among adolescents and young adults: a systematic review of their co-use. Clinical psychology review, 2012. 32(2): p. 105-121.

We could use separate variables for alcohol drinking and marijuana use, however, alcohol drinking and marijuana use is strictly prohibited by law in Bangladesh. So information on these is rare. Therefore, we have combined these two and named it ‘substance use’. Since we want to assess how psychosocial and environmental factors are associated with adolescents’ substances use (smoking, alcohol drinking and marijuana use), therefore we use these as outcomes. 

What forms of marijuana use were included?, since ‘drinking’ alcohol is specified, it would be good to maintain parallelism in the sentence.

Response: Thank you. Yes it would be nice to have similar structure of the questionnaire. However, we used a nationally representative secondary data (GSHS), so we have to use the questionnaire and the data as they collected. The questionnaire of the GSHS can be found in the following link:

https://www.cdc.gov/gshs/questionnaire/index.htm

Introduction

If it is known that 6.9% of adolescents are tobacco users, and 2% are alcohol users from 2017-2018 data, which is based on more recent data than the study under review, what is the justification for this study based on data from 2014? What is the gap that the current study fills in terms of assessing prevalence?

Response: Thank you for your query. Actually, the references (4 and 5, in earlier version of the manuscript) used for documenting the prevalence were published in 2018 and 2017; but they used data from Global Youth Tobacco Survey (GYTS), Bangladesh 2013, and the WHO STEPwise Surveillance (STEPS) data in 2010, respectively. And this current analysis is based on data collected in 2014, therefore, the prevalence reported in this current study is from the latest data. However, to avoid confusion we have rewritten the sentence as follows and used the original references: 

“The 2013 Global Youth Tobacco Survey of Bangladesh reported that 6.9% of adolescents (9.2% males and 2.8% females) are currently tobacco users. The prevalence of alcohol use across age groups is less than one percent (0.8%; men 1.5% and women 0.1%.”

Suggestion to include a description of the types of tobacco use in Bangladesh, as there may be many diverse forms of use. Line 99 – “limited to only tobacco (smoke or smokeless)”

Response: Thank you. We have added the following sentence in the first paragraph describing types of tobacco use in Bangladesh (Page 4, Lines 59-61)

“Around 35% of the total population in Bangladesh use either smoked cigarettes or bidis or smokeless tobacco products (tobacco flakes known as Zarda, betel leaf quid or paan, and gul”

What makes the authors believe that the reasons for adolescents using tobacco in global studies differ from the reasons why adolescents use in Bangladesh?

Response: Although there are some restrictions of smoking in public places and selling cigarettes to the minors; tobacco smoking or using other forms of tobacco is not prohibited by law in Bangladesh. Like many other countries, using tobacco, specifically smoking among adolescents is also socially and culturally unacceptable in Bangladeshi. The numbers of tobacco smokers are increasing rapidly because of the availability of cheap tobacco products, lack of strong tobacco control regulations, and weak enforcement of existing regulations. However, alcohol drinking and using marijuana is prohibited by law in Bangladesh. Since using these substances has enormous health impact among the adolescent, therefore more studies are required to identify whether the same factors contributes to substances use in Bangladesh and in other settings. Factors those are found to be associated with tobacco use in global studies might be true for Bangladesh. However, if data is available, it would be better to find out whether these factors that are actually associated or not with respective country’s data. It would help formulate appropriate culturally acceptable interventions. That’s why our study is important and has implication for Bangladesh.

Line 98-99 What is the reason why the authors want to include an urban sample on tobacco users? What proportion of Bangladesh is urban?

Response: Thank you. The proportion of urban population in Bangladesh is around 37% (According to statista in 2018; link below). Some earlier studies in Bangladesh did not use country representative sample, rather used only rural sample. Therefore, it is not possible to generalize the findings to the whole country. To get the whole picture of substance use, we need country representative data, not partial one. We, in our study, fill that gap by identifying factors associated with substance use using nationally representative sample which represents the whole country.

https://www.statista.com/statistics/761021/share-of-urban-population-bangladesh/

Line 100: What is the evidence that psychosocial factors are important? Perhaps the authors would like to define these factors, explain briefly how they are likely interlinked. In the paragraph on global studies (line 78-89) the authors mention many factors, but latel them interpersonal risk factors, protective factors, psychosocial distress, and so one is not sure what psychosocial factors entail exactly?

Response: Thank you. For addressing this issue we have added the following sentences in the text (Page 5 Line 92-101).

“There is a large body of literature that highlights psychosocial factors such as, loneliness, stress, depression, anxiety, hyperactivity/inattention, and low self-esteem as risk factors for substance abuse. For example, Page et. al. reported in a four country study (Philippines, China, Chile, and Namibia) that among adolescents often feeling lonely, worried, sad/hopeless, or having a suicide plan were associated with current smoking and drinking alcohol. Findings from a longitudinal study in Australia suggest that psychosocial risk factors, such as truancy, hyperactivity/inattention, and behavior problems, are associated with adolescent binge drinking and cannabis use. A recent review concludes that psychosocial risk factors of adolescent substance use are closely related to peer influence, and that parents tend to have the strongest effect on adolescents’ substance use behavior”

Line 106: MAE – Multiple adverse experiences – This construct is mentioned without any citation or background information. It seems like it is better suited as a measure rather than mentioned in passing at the end of the introduction.

Response: We have added following sentence in the introduction section justifying the inclusion of MAE here (Page 6 Lines 116-118).

“A growing number of studies document a graded relationship between adverse life circumstances and negative health outcomes, including health risk behaviors, such as substance use. Therefore, we examine to what extent multiple adverse experiences (MAE) affect tobacco and substance use behaviors among the adolescent population.”

We also have discussed briefly about the calculation procedure of MAE scores in our methodology section. Following paragraph is added to the text (Page 8-9, Lines 165-174)

“Calculation of multiple adverse experience (MAE) scores

We followed the Kaiser Permanente Childhood Adverse Experience (ACE) study to calculate the multiple adverse experience (MAE) score. We divided adolescents’ adverse experiences into two categories: psychosocial and socio-environmental factors. Details of these two categories are presented in Table 1. The MAE score for each adverse category was calculated by summing up the responses on the adverse experience questions. For each adverse experience, the response 0 meant the participant never or rarely experienced the adversity, while 1 meant the participant always or sometimes experienced that specific adversity. Higher MAE meant greater experience of adverse events. As for example, if an individual has experienced three adverse events, then the MAE score is 3.”

Data and Methods

Line 119: What is the disadvantage of using a school-based sample? 

Response: Thank you. The GSHS was conducted by WHO in collaboration with CDC. They have adopted this school-based methodology in collecting data. We are just using the secondary data of GSHS. There might be several disadvantages of using school-based sample which are reported in earlier studies. These includes: difficult to recruit school, increasing non-response rate, excluding students who are absent in the survey day (references below). With high nonresponse rates, the validity of population measurement is questionable. High nonresponse rate also increase the likelihood of sample bias because whole school may represent a specific segment of society. Moreover, while studying health risk behaviors using school-based sample, studies have documented that prevalence of health risk behaviors to be higher among absent respondents. These disadvantages might be considered while collecting sample. 

Hallfors D, Iritani B. Local and state school-based substance use surveys—availability, content, and quality. Eval Rev. 2002;26(4):418-437. 

White DA, Morris AJ, Hill KB, Bradnock G. Consent and schoolbased surveys. Br Dent J. 2007;202(12):715-717. 

Thorlindsson T, Bjarnason T, Sigfusdottir ID. Individual and community processes of social closure: a study of adolescent academic achievement and alcohol use. Acta Sociol. 2007;50: 161-178. 

Bovet P, Viswanathan B, Faeh D, Warren W. Comparison of smoking, drinking, and marijuana use between students present or absent on the day of a school-based survey. J Sch Health. 2006;76(4):133-137.

Michaud PA, Delbos-Piiot I, Narring F. Silent dropouts in health surveys: are nonrespondent absent teenagers different from those who participate in school-based surveys? J Adolesc Health. 1998;22(4):326-333.

However, to accommodate the disadvantage of using school-based sample, we have added the following limitation in study limitations in the discussion section (Page 15, Lines 337-339).

“Moreover, adolescent tobacco and substance use may be underreported, because the GSHS was restricted to adolescents who attended school and were present on the day of the survey.” 

What is the proportion of adolescents in Bangladesh who do not go to school and are using tobacco?

Response: According to Bangladesh Demographic Health Survey 2014, the proportion of the population that do not attends school is 18% (boys: 21.7%, girls: 15%) for children age 11-15 in Bangladesh. 

The 2013 Global Youth Tobacco Survey of Bangladesh reported that 6.9% of adolescents (9.2% males and 2.8% females) are currently tobacco users. 

However, we have included the information of the prevalence of tobacco use in the 1st paragraph of the introduction section.

How long was the self-administered questionnaire?

Response: The self-administered questionnaire was consisted of a total of 80 core, expanded and country-specific questions. The Bengali version of the questionnaire was used in the survey (Reference below).

Report of first Global School-based Student Health Survey (GSHS), Bangladesh, 2014. New Delhi: World Health Organization; 2018. World Health Organization, Regional Office for South-East Asia and National Centre for Control of Rheumatic Fever and Heart Disease, Ministry of Health & Family Welfare, Dhaka, Bangladesh Licence: CC BY-NC-SA 3.0 IGO.

What steps were taken by the study team to manage adolescent distress while answering questions on Multiple adverse experiences?

Response: To ensure privacy and confidentiality the survey was anonymous and voluntary.

Was student assent collected?

Response: Yes, it was. Informed consent from the students, parents, and/or school officials, as appropriate, was obtained.

We have already included this information in method section.

What language was the survey completed in and was it in the local language? 

Response: The Bengali version of the questionnaire was used in the survey (Reference below).

Report of first Global School-based Student Health Survey (GSHS), Bangladesh, 2014. New Delhi: World Health Organization; 2018. World Health Organization, Regional Office for South-East Asia and National Centre for Control of Rheumatic Fever and Heart Disease, Ministry of Health & Family Welfare, Dhaka, Bangladesh Licence: CC BY-NC-SA 3.0 IGO.

Do participating adolescents understand and use the term marijuana colloquially?

Response: Marijuana is not familiar word in Bangladesh. In Bengali it is well known as Ganja or weed.

Bangladesh specific item used to measure marijuana use in the GSHS: “During the past 30 days, how many times have you used marijuana (also called ganja or weed)?”

We have corrected it in the text.

In the creation of age-standardized weights, what specific marginal population proportions were used from the population and housing census, and why? Please provide this information in a supplemental table. Or is it just based on age?

Response: Thank you. We used total population of the age-group 11to18 years to standardize the prevalence. We did the direct method of age-adjusted prevalence to eliminate differences in crude prevalence in populations of interest that result from differences in the populations’ age distributions. However, it is usually done when we compare two or more populations at one point in time or one population at two or more points in time. Since we are using nationally representative data, and not comparing the prevalence with other sub-group of the population; we think this age-adjusted prevalence is not necessary for the current study. Therefore, we decided to remove the age-adjusted prevalence from the table as well as from the manuscript. We think deleting these prevalence would not affect the quality of the manuscript.

See Line 160-164: Multiple imputation will not likely make the sample more representative. The citation does not match the text preceding it and it is indistinct why multiple imputation was used. It would also be useful to explain why “ever had sexual intercourse” was included in this context: “To ensure representativeness and to prevent misinterpretation or any form of biases in the analysis, the allocation of missing values was done through a multiple imputation method using a logistic regression model by taking into account the known values, most frequently for “ever had sexual intercourse” [34].”

Response: Thank you very much. We apologize for such mistakes. We have corrected it. Since the response of the variable “sexual history” (‘ever had sexual intercourse’ in earlier version of the manuscript) contains 11.6% missing values, we imputed these missing values. That’s why we included sexual history there. We have corrected the sentence in the following way so that it would be clearly understand by the readers (Page 9, Lines 182-185).

“Imputation using a logistic regression model was used to estimate missing values from known values to account for missing data, most frequently for sexual history (i.e., for 11.6% of respondents). Age, gender, and school grade were included as covariates in the imputation.” 

The measures include no mention of MAE. On what basis were the measures selected? And why are they appropriate in Bangladesh? The adverse events description and combination of measures in Table 1 appear a bit arbitrary and not evidence-based. Suggestion to add a justification and description of how these measures were categorized and used based on the literature.

Response: Thank you. We have just followed the calculation procedure of the Adverse Childhood Experience (ACE). We just wanted to see to what extent the experience of one or more than one psychosocial or socio-environmental events are associated with substance use among adolescents. Therefore, we adopted the procedure of calculating ACE and calculated the MAE score for this study. We also inserted a paragraph describing the calculation procedure, which we indicated in responding a comment earlier. The same was followed in an earlier published article.

Khan, M.M.A., et al., Suicidal behavior among school-going adolescents in Bangladesh: findings of the global school-based student health survey. Social psychiatry and psychiatric epidemiology, 2020: p. 1-12. DOI: https://doi.org/10.1007/s00127-020-01867-z

Results

Line 175-176: Please clarify what is meant by “more than half of the adolescents had poor parental understanding?”

Response: Thank you. We have revised the sentence in the text.

Line 179-189: Are these differences significant, if so, please provide summary of test statistics (Confidence Intervals and p-values)?

Response: We have revised it in the text according to your suggestion.

Line 185 - factors than (those) who did not experience any (such factors).

Response: Thank you. We have corrected it in our revised manuscript.

Figure 1 does not show that boys are significantly more likely to use tobacco and substances. It only shows that the proportion of boys using tobacco is higher than girls, and only marginally so for substances.

Response: We have revised it accordingly.

How are the authors defining an adverse sexual history? The measures in Table 1 do not include this – it just asks, ‘Have you ever had sexual intercourse?’

Response: Sexual intercourse outside marriage is considered as great sin and socially unacceptable. Therefore, we defined ‘adverse sexual history’ by using the item from GSHS ‘ever had sexual intercourse’. However, the 1st reviewer suggested changing the variable name ‘adverse sexual history’ to ‘sexual history’, and we did it according to him in the revised manuscript. 

Line 209: The use of dose-response relationship appears inappropriate in this context as there is no experiment underway. The analysis shows an additive negative effect of adverse experiences.

Response: Thank you. We have changed the sentence as follows (Page 12, Lines 245-246):

“Experience of increased number of adverse psychosocial adversities was found to be associated with greater tobacco use in a graded manner”

Line 210-211: Why were both age and grade adjusted for as confounders?

Response: Thank you. While assessing the relationship of various socio-demographic, psychosocial and socio-environmental factors with tobacco and substance use using multivariable analysis, we checked the multicollinearity among the variables. We checked the value of Variance Inflation Factors (VIF), and found that in all cases the VIF values were less than 2. This indicates that multicollinearity is not a problem. Therefore, multivariable analyses were adjusted for all the socio-demographic variables included in this study. However, in some recent analyses (given below) using data from the GSHS, multivariable analyses were adjusted for both age and school grade. 

Tang, J.J., et al., Global risks of suicidal behaviours and being bullied and their association in adolescents: School-based health survey in 83 countries. EClinicalMedicine, 2020. 19: p. 100253.

Xi, B., et al., Tobacco use and second-hand smoke exposure in young adolescents aged 12–15 years: data from 68 low-income and middle-income countries. The Lancet Global Health, 2016. 4(11): p. e795-e805.

Tables 2 should provide sample N and counts in a column on unweighted data.

Response: We have revised Table 2 accordingly. Thank you.

Line 220: This seems like a sentence more appropriate as an explanation in the Discussion section.

Response: Thank you. We have deleted that sentence from the result section and the following sentence is added in the discussion section (Page 15, Lines 324-327) 

“However, we did not find statistically significant association between experience of multiple adverse socio-environmental events and adolescents’ substance use, which could be due to very few positive responses in the multiple adverse socio-environmental categories.”

Discussion

Line 227-229: Reference 26-28 are much more dated than prevalence rates reported in the Introductory paragraph.

Response: Although the references for prevalence of tobacco use in Bangladesh provided in the earlier version of the manuscript were published recently, they actually used data from 2013 GYTS or 2010 STEPS. We have changed those references to actual report published after those surveys in 2015 or 2011. Now those references are same.

Line 233: remove extra space before period

Response: Thank you. We have removed the extra space.

Line 267-270: It may be useful to bring this point earlier in the paper, maybe as a footnote to Table 1.

Response: Thank you for the suggestion. For the flow of the reading, we think those two sentences could be fine here as it is. 

Line 275- predictors is not an accurate description of this study’s findings

Response: Thank you. We have revised it throughout the manuscript.

Line 280-289 and line 242-252: The idea of parental involvement/ monitoring and understanding is repetitive and may be combined effectively.

Response: Thank you. We have deleted some repetitive/abundant explanations from the indicated lines.

Study strengths and implications should be described in greater detail.

Response: Thank you. We have rewritten the strengths of this study in the text and added the following paragraph describing the implications of this study (Page 16, Lines 344-355).

“Limitations notwithstanding, this study has the potential to inform the development and/or modification of prevention strategies. Teachers, parents or other school personnel could play important role in early identification of adolescents with these emotional and behavioral problems. However, they would require appropriate training (e.g. mental health training) and professional development opportunities to build their skills and self-efficacy to be able to recognize, and appropriately respond to the adolescents with these problems. Furthermore, the findings suggest the need for incorporating mental health education programs in school-curricula; and the importance of the family and social connections among these populations. Findings of this study underscore the importance of parental awareness to help and support adolescents in problems. Finally, since parental smoking increases the likelihood of adolescent smoking, universal prevention approaches which combine tobacco cessation campaign through mass media, including school- and community-based programs could be effective.”

Please review:

https://www.childwelfare.gov/topics/preventing/preventionmonth/resources/ace/

For other descriptions of adverse childhood events and categorization.

Response: Thank you very much. 

Line 314 – The authors themselves provide other prevalence estimates from Bangladesh, so it may seem prudent to avoid saying it is the first or even most recent?

Response: Thank you. We have revised it and re-written the conclusion section as follows (Page 16, Lines 357-366).

“This study, using a nationally representative data, estimates the prevalence and risk factors associated with tobacco and substance use among school-going adolescents in Bangladesh. The findings of this study have important implications for public health professionals and practitioners who work on adolescent substance abuse programs. Several adverse psychosocial (loneliness, anxiety, bullying, and sexual history), and socio-environmental factors (parental tobacco use, lack of peer support, and poor parental monitoring) were found to be significantly associated with adolescents’ tobacco and substance use. Although the relationships are naturally complex, some patterns are discernible. Further studies with longitudinal data are needed to determine the mechanisms, processes and directionality of these relationships.” 

It may be useful to discuss mental health promotion opportunities in light of psychosocial findings.

Response: Thank you. We have mention about the necessity of mental health in the implication section.

---

## [Decision Letter · Decision Letter 1]

30 Oct 2020

PONE-D-20-18742R1

Psychosocial and socio-environmental factors associated with adolescents’ tobacco and substance use in Bangladesh

PLOS ONE

Dear Dr. Rahman,

Thank you for submitting your manuscript to PLOS ONE. After careful consideration, we feel that it has merit but does not fully meet PLOS ONE’s publication criteria as it currently stands. Therefore, we invite you to submit a revised version of the manuscript that addresses the points raised during the review process.

Please make the editorial changes Reviewer 2 suggested.  ADEMIC EDITOR: Please insert comments here and delete this placeholder text when finished. Be sure to:

Indicate which changes you require for acceptance versus which changes you recommendAddress any conflicts between the reviews so that it's clear which advice the authors should followProvide specific feedback from your evaluation of the manuscript

Please make the editorial changes Reviewer 2 suggested. 

We look forward to receiving your revised manuscript.

Kind regards,

Stanton A. Glantz

Academic Editor

PLOS ONE

Reviewers' comments:

Reviewer's Responses to Questions

**Comments to the Author**

1. If the authors have adequately addressed your comments raised in a previous round of review and you feel that this manuscript is now acceptable for publication, you may indicate that here to bypass the “Comments to the Author” section, enter your conflict of interest statement in the “Confidential to Editor” section, and submit your "Accept" recommendation.

Reviewer #1: All comments have been addressed

Reviewer #2: All comments have been addressed

2. Is the manuscript technically sound, and do the data support the conclusions?

Reviewer #1: Yes

Reviewer #2: Yes

3. Has the statistical analysis been performed appropriately and rigorously? 

Reviewer #1: Yes

Reviewer #2: I Don't Know

4. Have the authors made all data underlying the findings in their manuscript fully available?

Reviewer #1: (No Response)

Reviewer #2: Yes

5. Is the manuscript presented in an intelligible fashion and written in standard English?

Reviewer #1: Yes

Reviewer #2: Yes

6. Review Comments to the Author

Reviewer #1: This was an interesting study using the latest interation of Global School-Based Student Health Survey in Bangladesh to examine association between childhood risk factors and tobacco use and substance use, which was a combined metric of alcohol and/or cannabis use. There was a big improvement from the last draft

Reviewer #2: There are some minor issues that remain in the manuscript, that the authors must address:

1. Below table 3, the authors still use the word predictors in the manuscript- "Model was adjusted for all the predictors included in this table."

2. Table 2 p-values and chi-square test values are presented in a slightly unusual format, with p-value at the base of each characteristic. Could the authors explore an alternative way of presenting whether participant characteristics are significant by tobacco users and substance users? Also, it would be useful if the authors would clarify that this table relates only to tobacco use and substance use and not tobacco use disorder or substance use disorder.

3. Use of the terms "substance abuse" remains in lines- 49, 85, 94, 121, 227, 360. If both use and abuse are being used in the manuscript, the authors may specify in the manuscript how these concepts differ and why they are both important for their study.

4. Suggestion to clarify in the manuscript - “Additionally, adolescents’ multiple adverse experience of psychosocial factors was

found to be associated with TU in a graded manner.” Perhaps the authors could say - For every additional adverse psychosocial experience there appeared to be an incremental association with increasing TU or it may work to give the exact increased odds of TU in the text. I believe the addition of MAE-relevant information in the manuscript after the initial review feedback is adequate, but in the abstract there needs to be greater clarity.

5. The rationale that alcohol and marijuana are strictly prohibited in Bangladesh does not appear to justify the creation of new substance use questions when other validated measures on substance use, including both alcohol and marijuana exist. For example, the Alcohol, Smoking and Substance Involvement Screening Test (ASSIST) developed by the World Health Organization (WHO) is widely used.

6. Please indicate if all 6 questions on alcohol use and all 4 questions on drug use from the GSHS were used. Please add a sentence on the use of the Bengali version of the questionnaire with its citation in the manuscript.

7. It may be suitable for the authors to acknowledge in the manuscript that there is no existing evidence to suggest that the reasons for adolescents using tobacco in Bangladesh are similar/different from the reasons why adolescents use in global studies.

8. Given that different tobacco prevalence studies captured varying distributions of urban-rural populations, the authors should mention what proportion of their study is an urban/ rural population in the manuscript.

9. It would help readers to understand the survey, if the authors would add details to the manuscript about the length of the self-administered questionnaire with references and mode (pen and pencil/ online).

10. As the survey asks adolescents about multiple adverse experiences, were there any events during survey administration where participant distress was voiced/reported/observed and what did authors do as a response? Was a counsellor/ psychosocial support available? The authors should add this information to the manuscript for the benefit of readers.

11. It is not clear whether student assent was collected in addition to informed consent. The authors may clarify in the manuscript.

12. Please add to the measures in the manuscript why 'adverse' was a word the authors used to describe sexual history in the Bangladeshi context.

7. PLOS authors have the option to publish the peer review history of their article (what does this mean?). If published, this will include your full peer review and any attached files.

Reviewer #1: No

Reviewer #2: No

---

## [Author Response · Author response to Decision Letter 1]

9 Nov 2020

November 9, 2020

Prof. Stanton A. Glantz

Academic Editor

PLOS ONE

Re: Manuscript Number PONE-D-20-18742R1

Dear Prof. Stanton A. Glantz,

Thank you very much for your kind consideration of our manuscript entitled, “Psychosocial and socio-environmental factors associated with adolescents’ tobacco and other substance use in Bangladesh”. We have revised the manuscript according to the comments by the reviewers. Their useful and productive comments helped us to improve the clarity and quality of the manuscript. Where we have changed the text, the corresponding sentences in the text have been highlighted with the track changes function. The page and line numbers referred to are those in the margins of our clean revised manuscript. Where textual changes are included in this response to reviewer, we have indented and italicized them, and removed references.

We hope that the revisions are satisfactory in addressing the issues raised by the editor and the reviewers, and look forward to hearing your decision about this article.

Yours sincerely,

Md. Mosfequr Rahman, PhD

Department of Population Science and Human Resource Development

University of Rajshahi

Rajshahi-6205, Bangladesh.

E-mail: mosfeque@ru.ac.bd

Reviewer #1

This was an interesting study using the latest interation of Global School-Based Student Health Survey in Bangladesh to examine association between childhood risk factors and tobacco use and substance use, which was a combined metric of alcohol and/or cannabis use. There was a big improvement from the last draft

Response: Thank you very much. 

Reviewer #2

There are some minor issues that remain in the manuscript, that the authors must address:

Response: Thank you very much. We have addressed the issues one by one in the following sections.

1. Below table 3, the authors still use the word predictors in the manuscript- "Model was adjusted for all the predictors included in this table."

Response: Thank you very much. We have revised it.

2. Table 2 p-values and chi-square test values are presented in a slightly unusual format, with p-value at the base of each characteristic. Could the authors explore an alternative way of presenting whether participant characteristics are significant by tobacco users and substance users? Also, it would be useful if the authors would clarify that this table relates only to tobacco use and substance use and not tobacco use disorder or substance use disorder.

Response: Thank you. We have revised Table according to your suggestion. We have mentioned in both the title of the table 2 and in the text that we are assessing the prevalence of tobacco use and other substance use among the adolescents, not the disorders.

3. Use of the terms "substance abuse" remains in lines- 49, 85, 94, 121, 227, 360. If both use and abuse are being used in the manuscript, the authors may specify in the manuscript how these concepts differ and why they are both important for their study.

Response: We had tried to be consistent in using the term ‘use’ throughout the manuscript; however, unintentionally in some places we missed to change it. We have revised it. However, the term ‘abuse’ in line number 89 (page line of the current version) is appeared as a risk factor of substance use; this ‘abuse’ is the experience of violence. So we keep it as it was. 

4. Suggestion to clarify in the manuscript - “Additionally, adolescents’ multiple adverse experience of psychosocial factors was

found to be associated with TU in a graded manner.” Perhaps the authors could say - For every additional adverse psychosocial experience there appeared to be an incremental association with increasing TU or it may work to give the exact increased odds of TU in the text. I believe the addition of MAE-relevant information in the manuscript after the initial review feedback is adequate, but in the abstract there needs to be greater clarity.

Response: Thank you. We have rewritten the mentioned line from the abstract as follows:

“Additionally, higher odds of tobacco use were observed among adolescents who reported 1 (AOR: 4.36 times; 95% CI: 1.34-14.24), 2 (AOR: 8.69 95% CI: 1.67-28.23), and ≥3 (AOR: 17.46; 95% CI: 6.20-49.23) adverse psychosocial experiences than who did not report any psychosocial events.”

5. The rationale that alcohol and marijuana are strictly prohibited in Bangladesh does not appear to justify the creation of new substance use questions when other validated measures on substance use, including both alcohol and marijuana exist. For example, the Alcohol, Smoking and Substance Involvement Screening Test (ASSIST) developed by the World Health Organization (WHO) is widely used.

Response: Thank you. To avoid confusion and conflict with other validated measures, and for better understanding of the readers we have change ‘substance use’ as ‘other substance use’ throughout the manuscript. Other substance use includes the use of alcohol and/or marijunana. 

6. Please indicate if all 6 questions on alcohol use and all 4 questions on drug use from the GSHS were used. Please add a sentence on the use of the Bengali version of the questionnaire with its citation in the manuscript.

Response: Thank you. We assessed the current use (in past 30 days) of tobacco (in the form of smoking or smokeless), alcohol and/or marijuana among the adolescents. Therefore, we only used those questions (2 for current tobacco user, 1 for current alcohol use, and 1 for marijuana use) which are given in the outcome section in the text (Page Lines,) to measure the current users of these substances. We have clearly mentioned it in the outcome section. 

We have added a sentence in method section (Page 7, Line 147) indicating that the Bengali version of the questionnaire was used in the survey with a reference. 

7. It may be suitable for the authors to acknowledge in the manuscript that there is no existing evidence to suggest that the reasons for adolescents using tobacco in Bangladesh are similar/different from the reasons why adolescents use in global studies.

Response: Thank you. We have added the following sentence in the introduction (Page 6, Line 117-119)

“Therefore, how psychosocial factors, which are found to be risk factors in some global studies, are associated with tobacco and other substance use among adolescents in Bangladesh is unknown.” 

8. Given that different tobacco prevalence studies captured varying distributions of urban-rural populations, the authors should mention what proportion of their study is an urban/ rural population in the manuscript.

Response: Thank you for your query. We use the secondary data from GSHS, Bangladesh, and there no information in GSHS on Rural-Urban sample. Data were not disaggregated by rural and urban. Therefore, we are unable to provide such information. However, the method section of GSHS only reported that it is a nationally representative survey. 

9. It would help readers to understand the survey, if the authors would add details to the manuscript about the length of the self-administered questionnaire with references and mode (pen and pencil/ online).

Response: Thank you. We have added the following sentences in the method section (Page , Lines )

“The questionnaire consisted of 80 core, expanded and country specific questions. The survey used the Bengali version of the questionnaire.”

10. As the survey asks adolescents about multiple adverse experiences, were there any events during survey administration where participant distress was voiced/reported/observed and what did authors do as a response? Was a counsellor/ psychosocial support available? The authors should add this information to the manuscript for the benefit of readers.

Response: Thank you. We also think this information would benefit the readers for better understanding the survey. Since we are using secondary data from the GSHS and no such information is available in the GSHS methodology. Therefore, we are unable to include this information in the manuscript.

11. It is not clear whether student assent was collected in addition to informed consent. The authors may clarify in the manuscript.

Response: Thank you. We have revised the sentence as follows (Page Line 150-11). And this is the information available in the GSHS methodology.

“To ensure voluntary participation, privacy, and confidentiality, informed consent was obtained from the students, parents and/or school officials”

12. Please add to the measures in the manuscript why 'adverse' was a word the authors used to describe sexual history in the Bangladeshi context.

Response: Thank you. For addressing this issue we have added the following sentences in the text (Page 5 Line 92-101).

Earlier studies documented that early sexual initiation (i.e., sexual initiation before 16 years old) has been associated with adverse sexual health outcomes at the time of first intercourse and also later in life with an increased risk of having multiple lifetime sexual partners, unprotected sex, acquiring sexually transmitted infections (STIs), unwanted pregnancy, and undesirable sexual outcomes, such as problems with orgasm and sexual arousal. Moreover, adolescents may not be equipped to manage the psychological consequences of sexual activity; experiencing regret and having a higher risk of depression. 

In addition, Bangladesh is a Muslim majority country and Islamic principles play a pivotal role in one’s life. Early and premarital sexual intercourse is not accepted and objectionable on social, religious, or moral grounds. A national survey reported that 13% of unmarried Bangladeshi male adolescents had sexual intercourse and much of this experience involves unprotected sexual intercourse with commercial sex workers. Moreover, religious, cultural and social norms are not in favor of the issue of sex education in schools, so Bangladeshi adolescents have very little knowledge on sexual and reproductive health. Bearing all these in mind, we use the term ‘averse’ while defining experience of sexual intercourse variable as ‘adverse sexual history’ in this study. However, it already creates confusion and another reviewer suggested to remove the term ‘adverse’. So we have deleted ‘adverse’. We also think global readers may get confused with the term “adverse”, so we have decided delete it. Therefore, we do not include any further description about it in the method section.

---

## [Editor Report · Decision Letter 2]

11 Nov 2020

Psychosocial and socio-environmental factors associated with adolescents’ tobacco and other substance use in Bangladesh

PONE-D-20-18742R2

Dear Dr. Rahman,

We’re pleased to inform you that your manuscript has been judged scientifically suitable for publication and will be formally accepted for publication once it meets all outstanding technical requirements.

Kind regards,

Stanton A. Glantz

Academic Editor

PLOS ONE
---

## [Editor Report · Acceptance letter]

13 Nov 2020

PONE-D-20-18742R2 

Psychosocial and socio-environmental factors associated with adolescents’ tobacco and other substance use in Bangladesh 

Dear Dr. Rahman:

I'm pleased to inform you that your manuscript has been deemed suitable for publication in PLOS ONE. Congratulations! Your manuscript is now with our production department. 

Kind regards, 

on behalf of

Professor Stanton A. Glantz 

Academic Editor

PLOS ONE